# A Novel Murine Multi-Hit Model of Perinatal Acute Diffuse White Matter Injury Recapitulates Major Features of Human Disease

**DOI:** 10.3390/biomedicines10112810

**Published:** 2022-11-04

**Authors:** Patricia Renz, Andreina Schoeberlein, Valérie Haesler, Theoni Maragkou, Daniel Surbek, Amanda Brosius Lutz

**Affiliations:** 1Department for BioMedical Research, University of Bern and Switzerland, 3010 Bern, Switzerland; 2Department of Obstetrics and Gynecology, Division of Feto-Maternal Medicine University Hospital, University of Bern, 3010 Bern, Switzerland; 3Institute of Pathology, University of Bern, 3010 Bern, Switzerland

**Keywords:** perinatal brain injury, diffuse injury, white matter injury, mouse model, gliosis, myelination failure, two-hit model

## Abstract

The selection of an appropriate animal model is key to the production of results with optimal relevance to human disease. Particularly in the case of perinatal brain injury, a dearth of affected human neonatal tissue available for research purposes increases the reliance on animal models for insight into disease mechanisms. Improvements in obstetric and neonatal care in the past 20 years have caused the pathologic hallmarks of perinatal white matter injury (WMI) to evolve away from cystic necrotic lesions and toward diffuse regions of reactive gliosis and persistent myelin disruption. Therefore, updated animal models are needed that recapitulate the key features of contemporary disease. Here, we report a murine model of acute diffuse perinatal WMI induced through a two-hit inflammatory–hypoxic injury paradigm. Consistent with diffuse human perinatal white matter injury (dWMI), our model did not show the formation of cystic lesions. Corresponding to cellular outcomes of dWMI, our injury protocol produced reactive microgliosis and astrogliosis, disrupted oligodendrocyte maturation, and disrupted myelination.. Functionally, we observed sensorimotor and cognitive deficits in affected mice. In conclusion, we report a novel murine model of dWMI that induces a pattern of brain injury mirroring multiple key aspects of the contemporary human clinical disease scenario.

## 1. Introduction

The WHO estimates that one in ten live births occurs preterm (prior to 37 weeks gestation) worldwide, with major regional differences [1]. While ongoing progress in neonatal care has improved the survival of preterm infants, increased survival in this population is accompanied by a climbing disease burden due to the chronic disabilities associated with prematurity [1]. Approximately 10% of preterm survivors suffer permanent motor impairment, while up to 25–50% of preterm infants exhibit other neurodevelopmental deficits, including cognitive, learning, and social-behavioral disabilities [1]. White matter injury is the predominant form of brain injury in survivors of preterm birth, resulting from preterm-birth-associated perinatal inflammatory and hypoxic–ischemic insults due to clinical scenarios, such as chorioamnionitis, neonatal sepsis, pulmonary immaturity, and perinatal hemodynamic instability [2]. Preclinical and clinical studies emphasize the importance of numerous perinatal hits in the etiology of WMI, with initial insults sensitizing the developing brain to subsequent injury [2,3]. 

Whereas large foci of necrosis were commonly identified in preterm white matter injury tissue prior to 2000, thanks to advancements in obstetric and neonatal care, diffuse non-necrotic lesions have become the predominant disease pathology in recent decades [4]. These diffuse lesions are best characterized histopathologically by the presence of reactive astrocytes and microglia, disrupted maturation of pre-oligodendrocytes, and consequent myelination failure [1]. While subsequent dysmaturational events lead to long-term abnormalities in brain development, including gray matter abnormalities, in the acute phase, cortical gray matter, as well as axons, are generally spared [1]. 

The key to understanding the pathophysiology underlying these diffuse white matter lesions is the maturation-dependent vulnerability of the oligodendrocyte lineage [5,6]. Susceptibility to preterm white matter injury peaks between 23 and 32 weeks postconception, a period during which the pre-oligodendrocyte (pre-OL), a pre-myelinating oligodendrocyte progenitor, predominates in human cerebral white matter. Compared with earlier and later developmental stages of the oligodendrocyte lineage, the pre-OL is highly vulnerable to oxidative stress [2,6]. The temporal appearance and spatial distribution of this cell type correlate with the magnitude and location of WMI in humans and in experimental studies [5]. Taken together, the evidence thus far generates a working hypothesis of disease pathogenesis in which multiple perinatal insults during the vulnerable phase of oligodendrocyte cell lineage development generate reactive glia and impair oligodendrocyte lineage maturation. The result is long-term myelination failure and dysmaturational events, culminating in a spectrum of motor, cognitive, and behavioral disabilities [1]. 

Despite years of study, the molecular underpinnings of white matter injury remain incompletely understood, therapeutic windows are poorly defined, and therapeutic options are extremely limited [1,3,7]. Given the evident paucity of human pathologic specimens from affected infant brains available for research, reliance on experimental models that reproduce key features of human disease is even greater than in other areas of study. Numerous mammalian models (mouse, rat, sheep, and non-human primate) have been developed to study white matter injury disease mechanisms and potential therapeutic interventions [8,9,10,11,12,13,14]. Many of these models have been developed to model the focal cystic necrosis produced by earlier WMI. Murine models that reproduce the two-hit etiology of WMI and produce lesions mimicking diffuse white matter injury are needed to ensure relevance to contemporary disease patterns and to allow the incorporation of molecular genetic tools established in mice. Here, we present a mouse model of acute diffuse perinatal white matter injury based on a two-hit hypoxic–inflammatory insult during the period of pre-OL predominance in murine white matter corresponding to 23–32 weeks postconception in humans [15]. Our model generates reactive changes in astrocytes and microglia, pre-OL maturation failure, defects in myelination, and behavioral deficits in the absence of cystic white matter lesions or extensive primary gray matter injury, thus recapitulating multiple key features of human acute diffuse WMI. 

## 2. Materials and Methods

### 2.1. Animals

All animal procedures were approved by the Veterinary Department of the Canton of Bern, Switzerland (Protocol reference number: BE19/85), and the animals were maintained under standard housing conditions.

### 2.2. Animal Model

C57BL/6 mouse pups on postnatal day (P) 2 weighing 1.5–1.8 g were randomly divided into injured and control group. Lipopolysaccharide (LPS; Escherichia coli strain O55:B5, Sigma Aldrich, St. Louis, MO, USA) was diluted in sterile saline to a working concentration of 0.2 mg/mL, and injected at a dose of 2 mg/kg subcutaneously (s.c.) between the scapulae. Healthy control pups received an equivalent weight-adjusted saline injection. Following injection, pups were returned to their home cages. Hypoxia (8% O_2_/92% N_2_, 3 L/min) was performed 6 h after LPS injection in a temperature-controlled isolette (34 °C) for 25 min under continuous O_2_ monitoring. Healthy control pups were removed from dams and exposed to room air during this time (Figure 1). LPS dose and hypoxia duration were titrated to optimize for significant myelination defects and minimal animal mortality. 

### 2.3. Tissue Preparation

Mice pups at P3 (1 day post injury, 1 dpi) and P4 (2 days post injury, 2 dpi) were euthanized using rapid decapitation. Brains were removed and directly fixed in 4% paraformaldehyde (PFA) for 20 h at 4 °C. Mice pups at P11 (9 days post injury, 9 dpi) were euthanized with a terminal dose of sodium pentobarbital (150 mg/kg body weight, i.p.; Esconarkon, Streuli Tiergesundheit AG, Switzerland) and transcardially perfused with phosphate-buffered saline (PBS). Brains were fixed in 4% PFA for 22 h at 4 °C. Following fixation, brains were transferred to PBS. Following removal of the cerebelli and olfactory bulbs, brains were dehydrated through sequential emersion in ethanol and xylene and embedded in paraffin. Brains were sectioned into 6 µM slices using an HM 340E rotary microtome (Thermo Fisher Scientific, Waltham, MA, USA). Brains were sectioned in the coronal plane at the level of the hippocampus corresponding with provided illustrations. This plane was chosen because it allows analysis of the corpus callosum, as well as the internal capsule, regions typically affected in human WMI. Sections outside of this plane or that exhibited lost tissue integrity were excluded from our analysis (Appendix A).

### 2.4. Hematoxylin and Eosin (H and E) Staining

Paraffin-embedded tissue sections were deparaffinized and rehydrated. Tissue sections were then immersed in Mayer’s hematoxylin (Sigma Aldrich) for 12 min, washed with tap water for 2 min, immersed in eosin (Sigma Aldrich) for 1 min, and washed quickly in 95% ethanol. Tissue sections were then dehydrated by immersion in serial baths of increasing ethanol concentrations, followed by clearing in xylene twice and mounting using Eukitt. 

### 2.5. Histologic Analysis

Paraffin-embedded H-and-E-stained control and injured brains were evaluated by a blinded neuropathologist for the presence of macro- and microscopic cystic lesions (microcysts, previously defined as focal lesions with a diameter of approximately 1 mm) [16,17]. At least 2 brain sections per animal were examined.

### 2.6. Immunohistochemistry

Paraffin-embedded tissue sections were deparaffinized and rehydrated. Sections were then heated in a pressure cooker in 0.1 M sodium citrate buffer for 12 min for antigen retrieval. Sections were washed with Tris-buffered saline (TBS) Tween^®^ 20 (Sigma Aldrich) and blocked with 10% goat serum and 1% bovine serum albumin (BSA) (Sigma Aldrich) in TBS for 1 h at room temperature. Subsequently, sections were incubated overnight at 4 °C with primary antibodies against the following proteins: Olig2 (1:200, ab109186, Abcam, Cambridge, UK), Ki67 (1:100, 550609, BD, Pharmingen, BD Biosciences, Franklin Lakes, NJ, USA), CNPase (1:200, c5922, Sigma Aldrich), NG2 (ab5320, 1:100, Milipore, Merck KGaA, Darmstadt, Germany), MBP (1:200, ab40390, Abcam), Iba1 (1:2000, ab178846, Abcam), GFAP (1:500, mab360, Millipore), and NF200 (1:1000, smi31r100, Novus Biologicals, Centennial, CO, USA). After washing, the sections were incubated with the appropriate secondary antibody, either Alexa fluor^®^ 488-conjugated or Alexa fluor^®^ 594-conjugated (Thermo Fisher Scientific), for 1 h at room temperature in the dark. The tissue was counterstained with 4ʹ,6-diamidino-2-phenylindole (DAPI) (Sigma Aldrich). NF200 stains were incubated with peroxidase-labeled secondary antibody (1:100; DAKO, Glostrup, Denmark), followed by diaminobenzamidine and EnVision+ System HRP (DAKO) for visualization. The tissue was counterstained with hematoxylin. Cell death was quantified using a terminal deoxynucleotidyl transferase dUTP nick-end labeling (TUNEL) in situ cell death detection kit (TMR rot, Sigma Aldrich). Blocked sections were stained according to the manufacturer’s protocol (Chapter 3.3.4, Version 17, Sigma-Aldrich). After TUNEL staining, sections were stained with primary antibodies against the oligodendrocyte marker Olig2 (diluted in 10% goat serum and 1% BSA in PBS; incubated overnight at 4 °C). Sections were then incubated with Alexa fluor^®^ 488-conjugated secondary antibody (Thermo Fisher Scientific) for 1 h at room temperature and then counterstained with DAPI.

### 2.7. Image Analysis

All quantifications were carried out blinded to the experimental group. Images were either acquired with a DM6000 B microscope (Leica Microsystems, Wetzlar, Germany) or scanned using a Panoramic 250 Flash II slide scanner (3DHISTECH, Budapest, Hungary). The regions of the corpus callosum, globus pallidus and internal capsule, and cerebral cortex were outlined. For quantification of Ki67, TUNEL, CNPase, and NG2, cells were counted manually within the region of the corpus callosum. For MBP and GFAP, staining was quantified using ImageJ Software v1.47 (Rasband, W.S., National Institutes of Health, Bethesda, MD, USA, http://imagej.nih.gov/ij (accessed on 4 February 2022). Briefly, using a custom macro, the signal area within the brain region of interest and above a defined signal threshold was quantified as the percentage of the defined area. For Iba1 staining, cells were counted manually in each region of interest. The reactivity index was quantified in a multistep process according to [18]. Briefly, using a custom Image J macro, we first determined a signal area at a signal threshold of 40 (approximation for the whole cell area). Subsequently, a second signal area (approximation for the cell body area) was determined at a signal threshold of 90, a minimum size of 150 pixels, and a circularity index of 0.1–1. Cell body fraction was defined as the cell body area divided by the whole cell area. For IHC experiments, two brain sections were quantified and the results averaged to generate each datapoint. In rare cases, we were only able to quantify IHC results for one brain section because of insufficient tissue preservation.

### 2.8. Hindlimb Foot Angle

Foot angle was measured in video recordings of walking pups (13 days post injury, 13 dpi) by drawing a line from the center of the heel through the middle (longest) toe. The measurement was only performed when a pup took a full step in a straight line, and both hindfeet were flat on the ground. Three to five foot angles were measured, and the average angle was calculated for each pup [19].

### 2.9. Rotarod

Motor coordination and balance were determined in a rotarod test for injured and healthy control mice (28 days post injury, 28 dpi). Mice were trained on one day to remain on a rotating rod (five rotations per minute (rpm)) over a five-minute period. The following day, the rotational speed was increased continuously from 15 to 33 rpm, alternating between forward and reverse rotation modes. Trials were terminated when the mice fell off the rotarod, clung to the bar for two full rotations, or remained on the bar for five minutes. The procedure was repeated for a total of three trials, each separated by 15 min. The mean latency to fall was analyzed.

### 2.10. Novel Object Recognition

A novel object recognition task was used to assess recognition memory at 28 dpi [20]. Mice were placed in an open field where two identical objects were placed. Four hours later, one object was replaced by a new one at the same position, and the mice explored the open arena in the presence of both the new and a familiar object. The total time spent exploring the objects was limited to 20 s, with a maximum time of 10 min. Object types were placed on a randomly assigned side (left or right) of the open field to eliminate any possible side or object preference. Novel object recognition was demonstrated by discrimination ratios (novel object interaction/total interaction with both objects) above 0.5.

### 2.11. Statistical Analysis 

Given the sample size < 30 for all of our analyses and, therefore, the inability to reliably test for normal distribution of data, we applied the Mann–Whitney test for the pairwise comparison of continuous variables throughout the manuscript. Significance was determined at *p* < 0.05.

## 3. Results

### 3.1. Perinatal Inflammation–Hypoxia Does Not Lead to the Formation of Macroscopic or Microscopic Cystic Lesions

Contemporary cohorts of human postmortem tissue affected by perinatal white matter injury reveal a strong decline in the burden of necrotic lesions in periventricular white matter tracts in comparison with older cohorts [1]. We, therefore, began with a histologic evaluation of H-and-E-stained paraffin sections from our mouse model. This analysis, performed blinded at 1 dpi, 2 dpi, and 9 dpi, did not detect the presence of matrix changes corresponding to cystic lesions in the gray or white matter of any control or injured brains at the timepoints examined (Figure 2A). In contrast to earlier disease cohorts, studies of modern human perinatal white matter injury demonstrate only minimal neuronal loss in the cortical and deep gray matter regions [17]. When we quantified cell death in the cerebral cortex using TUNEL staining, we observed a transient significant increase in cortical cell death at 1 dpi. We did not observe this increase at 2 and 9 dpi (Figure 2B).

### 3.2. Perinatal Inflammation–Hypoxia Insult Induces White Matter Hypomyelination

A defining feature of perinatal white matter injury is white matter hypomyelination [1,21]. We evaluated myelination using immunohistochemistry for myelin basic protein (MBP) at 9 dpi in three brain regions: the corpus callosum, the cerebral cortex, and the region of the internal capsule and globus pallidus (Figure 3A). These experiments revealed a significant decrease in MBP immunoreactivity in the corpus callosum region and overlying cortex (Figure 3B,D). Myelination remained unchanged in the internal capsule and globus pallidus region (Figure 3D). Quantification of axon density in the corpus callosum at 9 dpi using immunohistochemistry for neurofilament (NF200) did not reveal any significant differences between injured and control tissue (Appendix A). 

### 3.3. Perinatal Inflammation–Hypoxia Leads to Impaired Oligodendrocyte Lineage Differentiation

Studies of human postmortem tissue affected by WMI have revealed that impaired oligodendrocyte maturation rather than a loss of mature oligodendrocytes, underlies white matter hypomyelination in this disease [2,4,6,16]. We, therefore, sought to understand the cellular dynamics underlying decreased myelination in the corpus callosum 9 days after injury in our disease model. We began by quantifying cell proliferation and cell death within the oligodendrocyte lineage in this region using Olig2 (oligodendrocyte lineage marker) immunohistochemistry combined with Ki67 (cell proliferation marker) immunostaining and TUNEL (cell death marker) labeling. These experiments indicated a significant decrease in Olig2+ cell proliferation at 1 and 2 dpi and a significant increase in Olig2+ cell death at 1 and 9 dpi (Figure 4A–D). Examination of the actual numbers of cells dying revealed an approximate 10–15% decrease in the cell proliferation of Olig2+ cells in the injured versus control corpus callosum samples and a rate of Olig2+ cell death in this region of approximately 3–5% in comparison with approximately 2% in healthy control mice. Despite these differences, the total number of NG2+ oligodendrocyte precursor cells, as well as the total number of Olig2+ cells, remained unchanged at these timepoints according to our analysis (Figure 4E–H). When we examined oligodendrocyte lineage differentiation using immunohistochemical labeling for CNPase, a marker of oligodendrocyte maturation beyond the pre-OL stage, we observed a significant decrease in the percentage of Olig2+ cells in the corpus callosum expressing CNPase in injured mice versus control mice, suggesting impaired oligodendrocyte lineage maturation (Figure 4G–I).

### 3.4. Perinatal Inflammation–Hypoxia Induces Reactive Changes in Microglia and Astrocytes

Another hallmark of human perinatal white matter injury is reactive changes in microglia and astrocytes [4,17,21]. We examined this feature in our disease model, for the corpus callosum, the cerebral cortex, and the region of the internal capsule and globus pallidus. Using immunohistochemistry for Iba1, we quantified the density of Iba1+ cells in each region of interest, as well as the fraction of the total microglia cell area (Iba1+ area) occupied by the cell body (cell body fraction) as a proxy for microglial reactivity. This analysis revealed an initial significant increase in the number of microglia occupying the corpus callosum and cerebral cortex regions of injured mice occurring within 24 h of injury (Figure 5B,E). This increase in cell number was no longer significant in comparison with control mice one day later in the corpus callosum but persisted in the cerebral cortex at 2 dpi (Figure 5E). The globus pallidus and internal capsule region showed no significant increase in microglia number in injured mice in comparison with control mice at the timepoints examined (Figure 5E).

When we examined the cell body fraction of Iba1-immunoreactive cells in injured mice, we noted a rapid increase in cell body fractions in the cerebral cortex, as well as the globus pallidus and internal capsule region, that were significant in comparison with control mice (Figure 5E). The cell body fraction of microglia in the corpus callosum did not exhibit a significant increase upon injury. Instead, microglia from both control and injured brains in this region displayed increased cell body fractions. The magnitude of the cell body fraction for microglia in the control corpus callosum was comparable to that observed for the microglia of the injured cerebral cortex and globus pallidus and internal capsule at 1 dpi (Figure 5E). When we examined cell morphology, the microglia from the corpus callosum of control brains appeared amoeboid, with few and short cellular processes (Figure 5C). By 9 dpi, microglia exhibited a mature ramified resting morphology and, therefore, a low cell body fraction across all the brain regions examined in both control and injured mice (Figure 5E). 

We quantified the percentage area occupied by GFAP immunoreactivity as a measure of astrocyte reactivity. This analysis revealed a significant increase in astrocyte reactivity in injured compared with control mice in all three brain regions examined at 1 dpi (Figure 5A,D). This difference was no longer significant at 2 dpi in the corpus callosum but persisted at 2 dpi in the cerebral cortex, as well as the globus pallidus and internal capsule region (Figure 5D).

### 3.5. Perinatal Inflammation–Hypoxia Results in Sensorimotor and Cognitive Deficits 

Perinatal white matter manifests as a broad spectrum of motor, cognitive, and behavioral deficits in affected individuals [1]. We aimed to test whether our disease model also yielded motor and cognitive deficits using a neonatal locomotion assay, a rotarod test, and a novel object recognition test [19,20,22]. The neonatal locomotion assay analysis revealed a significant increase in ambulation angle in pups at 13 dpi subjected to our disease model in comparison with control pups, indicating impaired sensorimotor development (Figure 6A). The rotarod test analysis demonstrated that the motor performances of the injured mice were significantly impaired compared with the performances of control mice at 28 dpi (Figure 6B). In addition, novel object testing revealed impaired recognition memory in injured mice (Figure 6C). 

## 4. Discussion

### 4.1. Perinatal Inflammation and Hypoxia Produce Diffuse Perinatal White Matter Injury

Two main etiologies of acute perinatal white matter injury are recognized: hypoxia-ischemia and inflammation [1]. White matter injury results when these insults occur in regions where the oligodendrocyte lineage is predominated by pre-OL [2]. Clinically, hypoxic insults may occur in the context of placental insufficiency, birth asphyxia, impaired cerebral autoregulation, and neonatal pulmonary insufficiency. Inflammatory insults may be caused by perinatal clinical scenarios, including chorioamnionitis and neonatal sepsis. Animal models of perinatal white matter injury have been developed across various species. While large-animal models more closely model human physiology and offer the potential for instrumentation, rodent models continue to offer key advantages in terms of time, cost, and ease of manipulation. Models driven by neuroinflammation achieve myelination defects following single or repeated administrations of inflammatory stimuli, including pro-inflammatory cytokines, bacterial-mimicking LPS, viral-mimicking Poly I:C, or live infectious organisms, during the window of white matter vulnerability [23,24,25]. The most well-known hypoxia–ischemia-driven model of perinatal white matter injury is the classical Rice–Vanucci model, which achieves brain injury through a hypoxic–ischemic insult in postnatal day 7 neonatal rat pups by combining unilateral carotid artery ligation with an episode of systemic hypoxia [26]. In this model, the combination of hypoxia and ischemia is required to produce brain injury. Rats are able to survive periods of hypoxia without brain damage, and the unilateral induction of ischemia through carotid artery ligation can be compensated for by contralateral circulation thanks to the rats’ complete Circle of Willis [27]. This model has been modified in recent years to optimize injury timing relative to the time course of oligodendrocyte lineage development in rodent white matter [27]. Steps to adapt this model to mice have revealed significant strain-specific differences in sensitivity to hypoxic–ischemic injury [28]. The injuries produced in these hypoxia–ischemia models often result in cystic white matter lesions that are not representative of modern WMI pathology [23]. Whether a common mechanism leads to myelination defects in models driven by both disease etiologies (inflammation v. hypoxia–ischemia) remains unclear. 

Evidence from human and animal studies has demonstrated that inflammation potentiates the effects of subsequent brain insults and that WMI often results following multiple perinatal insults [24,26,29,30,31,32]. This notion of increased susceptibility to brain injury following a sensitizing perinatal insult has obvious clinical implications. Increased ability to recognize the occurrence of a first hit may represent an opportunity for intervention to protect the brain from or prevent the occurrence of a second insult [3]. Recapitulating a two-hit disease etiology in animal models employed for the study of disease mechanisms and therapeutic interventions is important to increase the likelihood of successful translation. 

The current model is a multi-hit injury paradigm involving two successive systemic acute insults, one inflammatory and one hypoxic. Despite involving two distinct insults, the severity of the resulting injury was confined to diffuse changes in glial reactivity and myelination outcomes rather than the production of cystic lesions. 

### 4.2. Impaired Oligodendrocyte Differentiation Appears to Underlie Perinatal Hypoxia–Inflammation-Induced Myelination Defects

Hypomyelination in cerebral white matter is a requisite finding for the diagnosis of perinatal white matter injury [1]. In humans, the corpus callosum and internal capsule regions are particularly affected [17]. We observed significant reductions in myelination in the corpus callosum and cortex at 9 dpi in our disease model. In contrast, we found no significant decrease in myelination, as evaluated by MBP immunoreactivity, in the internal capsule and globus pallidus region, although this region displayed a robust induction of microglia and astrocyte reactivity. Reductions in cortical myelin in human white matter injury have not been well-characterized. The lack of a myelination phenotype in the internal capsule region of our injured brains could be due to region-specific differences in susceptibility to injury in humans versus rodents [33]. Alternatively, there may be defects in myelination in this region that require other methods (e.g., electron microscopy) of detection. 

Years of research have led to the consensus that the myelination defects seen in perinatal white matter injury stem from impaired maturation of the oligodendrocyte lineage rather than the death of mature oligodendrocytes [5,6,16]. In line with these findings, we found evidence for impaired maturation of the oligodendrocyte lineage in the corpus callosum at 9 dpi. The acute response of the oligodendrocyte lineage to injury has also been studied in human tissue affected by WMI, revealing initial rapid degeneration of the pre-OL population followed by replenishment of this cell population through a phase of increased oligodendrocyte precursor proliferation [6,16,34,35]. The blocked differentiation of this regenerated pre-OL pool underlies the ensuing persistent myelination defects [16]. We examined the death and proliferation of Olig2+ cells at 24 and 48 h after injury. The increase in Olig2+ cell death that we observed (2–5%) is very modest in comparison to the magnitude of cell death observed at early timepoints in human tissue [6,35]. Instead of increased Olig2+ cell proliferation, we observed a significant decrease in Olig2+ cell proliferation at 1 and 2 dpi. Interestingly, these cellular dynamics were not pronounced enough to significantly affect the total number of OPCs or Olig2+ cells at the timepoints examined, suggesting that they were relatively small and transient phenomena. One possibility is that an initial wave of cell death and compensatory proliferation already occurred by 24 h after injury or that OPC proliferation occurred between the 2 and 9 dpi timepoints. Indeed, experimental models have suggested a quick rise in pre-OL degeneration peaking within the first 12 h after injury and a phase of OPC proliferation several days after injury [6,31,36]. Alternatively, it could be that pre-OL death is more pronounced in the severe form of perinatal white matter injury involving tissue necrosis, cystic lesion formation, and significant neuronal cell death and that our failure to observe this phenomenon reflected the milder nature of our insult. At 9 dpi, NF200 immunohistochemistry did not reveal axonal loss that could underlie the observed myelination defects. To fully investigate a role for secondary demyelination, axon studies should be performed at later timepoints and include specific markers of axon pathology. 

### 4.3. Observed Reactive Changes in Microglia and Astrocytes after Perinatal Inflammation–Hypoxia Appear to Be Transient 

In human tissue, the duration of reactive gliosis and blocked oligodendrocyte maturation remains somewhat unresolved due to a dearth of neuropathological data from human infants. Nonetheless, the scant evidence that has been gathered suggests that these features persist for at least months after term-equivalent age [4,17]. 

In line with studies of human postmortem brain tissue affected by WMI, we observed evidence of microglia and astrocyte reactivity in our disease model. These changes, determined by an increased GFAP+ signal area and an increased Iba1+ microglia cell body fraction relative to control animals, were no longer detectable by 9 dpi, suggesting that reactive gliosis was transient in our model. The specific gene expression changes characteristic of the reactive astrocytes and microglia that form in WMI are incompletely understood [37,38,39]. Evaluation of these changes was beyond the scope of our study. Unveiling the molecular identity and functional properties of these cells is a fascinating topic for future study. Our results may also reflect the more robust regenerative capacity of mice in comparison with humans [40]. It is also possible that additional postnatal insults might contribute to sustaining gliosis in human infants with white matter injury. 

### 4.4. Glial Reactivity in Developing Versus Mature Brains

Glial reactivity has, to date, mostly been studied in mature brains. We are still learning about differences in the capacity of microglia and astrocytes to respond to injury and the nature of these responses in perinatal versus adult brains [37]. Whether transient physiologic reactive changes in microglia and astrocytes are a part of normal development is not fully understood. Proliferative microglia of the developing white matter are known to exhibit an amoeboid reactive-like morphology under physiological conditions [37]. Recent single-cell-RNA-sequencing studies have identified molecular markers of this population, the so-called proliferation-associated microglia, or PAMs, and have highlighted similarities in gene expression with disease-associated microglia (DAMs) in the adult brain [41,42,43]. We observed an increased cell body fraction in microglia in control corpus callosum samples in our disease model at 1 and 2 dpi, potentially reflecting the presence of this PAM population. The effect of this microglia subtype on astrocyte reactivity in developing white matter has not yet been investigated. Furthermore, the relevance of these cells to human WMI and how this microglial population influences the response of white matter to perinatal insults is an area of active investigation. 

## 5. Conclusions

The identification of animal models that faithfully reproduce key features of human disease is essential to the advancement of research on these diseases, especially those diseases for which human tissue is difficult to procure. Even with improved animal models, no single model recapitulates all aspects of human pathophysiology. Models must be carefully chosen for the research question at hand and validated, when possible, with postmortem tissue. The use of human cellular models of white matter injury offers an exciting model system complementary to the use of animal models for furthering our understanding of this disease and confirming the relevance of findings in these animal models to human physiology [44]. We propose a mouse model of acute diffuse perinatal white matter injury that produces myelination defects, impaired oligodendrocyte maturation, reactive gliosis, and motor and cognitive deficits, allcharacteristics of contemporary human white matter injury. Limitations of our study include a histologic analysis up to only 9 days after injury, limited power of statistical analyses due to the low sample number, and missing molecular characterizations of astrocytes and microglia. Our model did not result in the formation of cystic white matter lesions, making it amenable to future studies assessing the pathophysiology of and novel therapeutic approaches for modern perinatal white matter injury. 

## Figures and Tables

**Figure 1 biomedicines-10-02810-f001:**
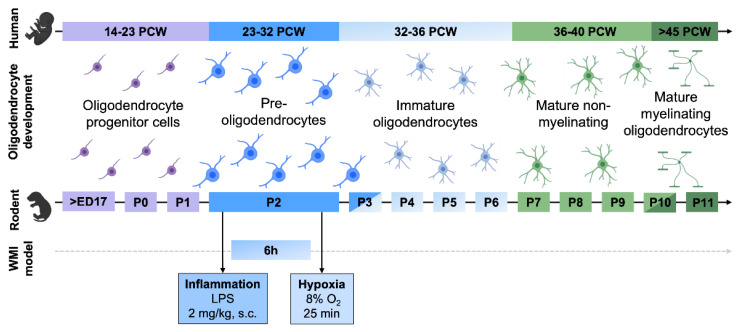
Murine 2-hit inflammatory–hypoxic model for acute diffuse perinatal white matter injury. WMI was induced in postnatal day 2 (P2) mice pups by s.c. injection of LPS, followed by exposure to hypoxia for 25 min 6 h later. PCW, postconception week; ED, embryonic day; P, postnatal day. Created with BioRender.com with data from [8].

**Figure 2 biomedicines-10-02810-f002:**
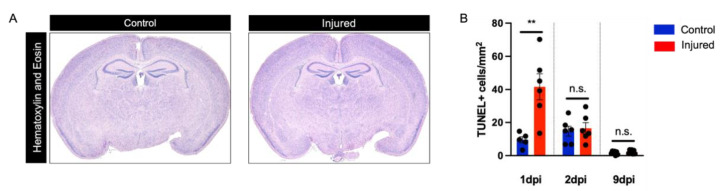
Evaluation of cystic lesion formation and cortical cell death. (**A**) Representative images of hematoxylin and eosin staining of control and injured brains at 9 dpi. (**B**) Quantification of cell death in mouse cortex using TUNEL staining in control and injured mice at 1 dpi (U = 1, *p* = 0.0087), 2 dpi (U = 18, *p* >0.999), and 9 dpi (U = 18, *p* = 0.4557). Sample amounts: 1 dpi: control *n* = 5, injured *n* = 6; 2 dpi: control *n* = 6, injured *n* = 6; 9 dpi: control *n* = 11, injured *n* = 9. Data are presented as mean ± SEM; n.s., not significant; ** *p* < 0.01.

**Figure 3 biomedicines-10-02810-f003:**
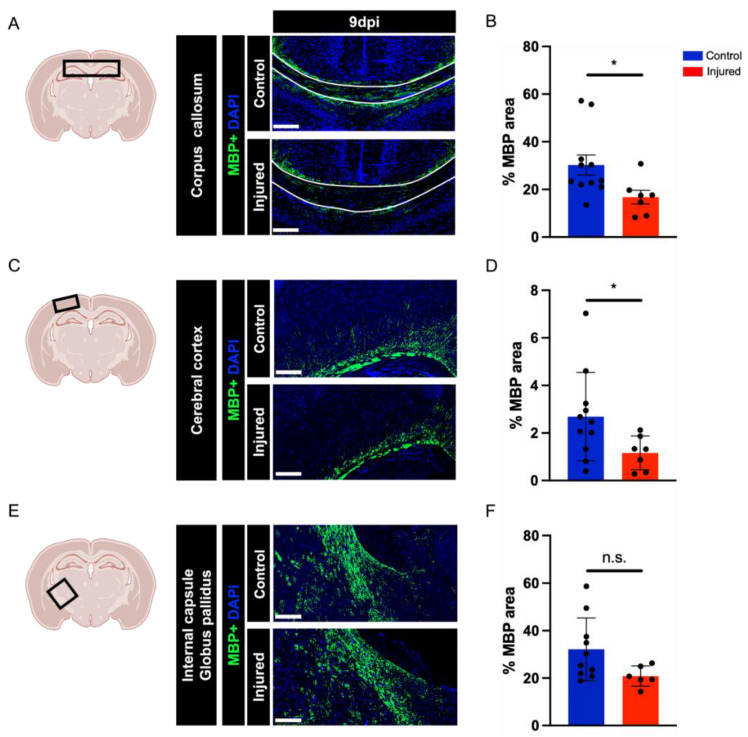
Quantification of myelination defects. Coronal brain sections and representative images indicating the brain regions used for quantification: corpus callosum (**A**), cerebral cortex (**C**), and internal capsule and globus pallidus (**E**). Quantification of MBP immunohistochemistry of injured and control brains at 9 dpi in the corpus callosum (U = 2, *p* = 0.0347) (**B**), cerebral cortex (U = 7, *p* = 0.0513) (**D**), and internal capsule and globus pallidus (U = 1, *p* = 0.0159) (**F**). Scale bar: 200 µm. Sample amounts: 9 dpi: control *n* = 11, injured *n* = 7. Data are presented as mean ± SEM; n.s., not significant; * *p* < 0.05.

**Figure 4 biomedicines-10-02810-f004:**
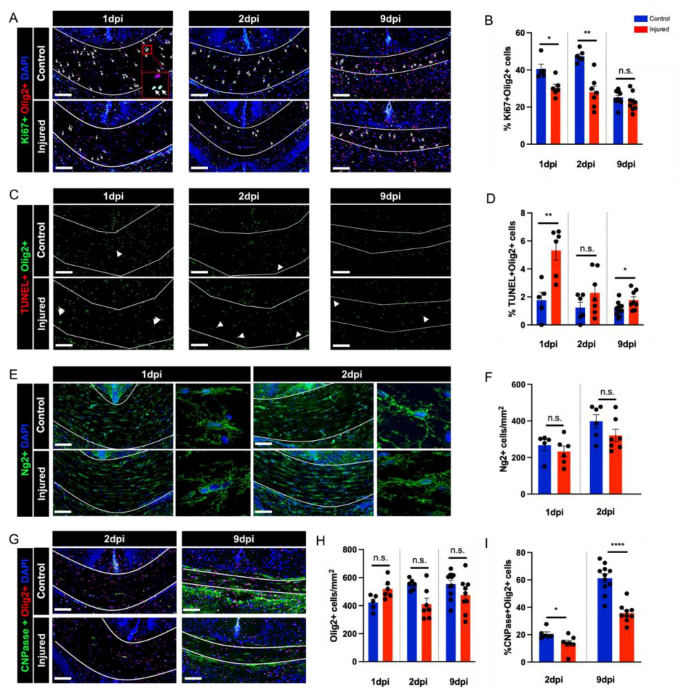
Oligodendrocyte cell lineage dynamics underlying myelination failure. (**A**,**B**) Representative images and quantification of oligodendrocyte cell lineage proliferation (Olig2+Ki67+) in the corpus callosum at 1 dpi (U = 2, *p* = 0.0173), 2 dpi (U = 0, *p* = 0.0012), and 9 dpi (U = 31, *p* = 0.1754). Arrows indicate double staining for oligodendrocyte marker Olig2 (red) and proliferation marker Ki67 (green). (**C**,**D**) Representative images and quantification of oligodendrocyte cell lineage death (Olig2+TUNEL+) at 1 dpi (U = 1, *p* = 0.0087), 2 dpi (U = 14, *p* = 0.3450), and 9 dpi (U = 17, *p* = 0.0259). Arrows indicate double staining for Olig2 (green) and TUNEL staining for DNA fragmentation (red). (**E**,**F**) Representative images and quantification of oligodendrocyte precursor (OPC) cell numbers (NG2+) at 1 dpi (U = 9, *p* = 0.3290) and 2 dpi (U = 10, *p* = 0.1305). (**G**–**I**) Representative images and quantification of oligodendrocyte maturation (Olig2+ and Olig2+CNPase+) at 2 dpi (U = 2, *p* <0.0001) and 9 dpi (U = 6, *p* = 0.0350). Scale bar: 100 µm. Sample amounts: 1 dpi: control *n* = 5, injured *n* = 6; 2 dpi: control *n* = 6, injured *n* = 7; 9 dpi: control *n* = 11, injured *n* = 9. Data are presented as mean ± SEM; n.s., not significant; * *p* < 0.05, ** *p* < 0.01, and **** *p* < 0.0001.

**Figure 5 biomedicines-10-02810-f005:**
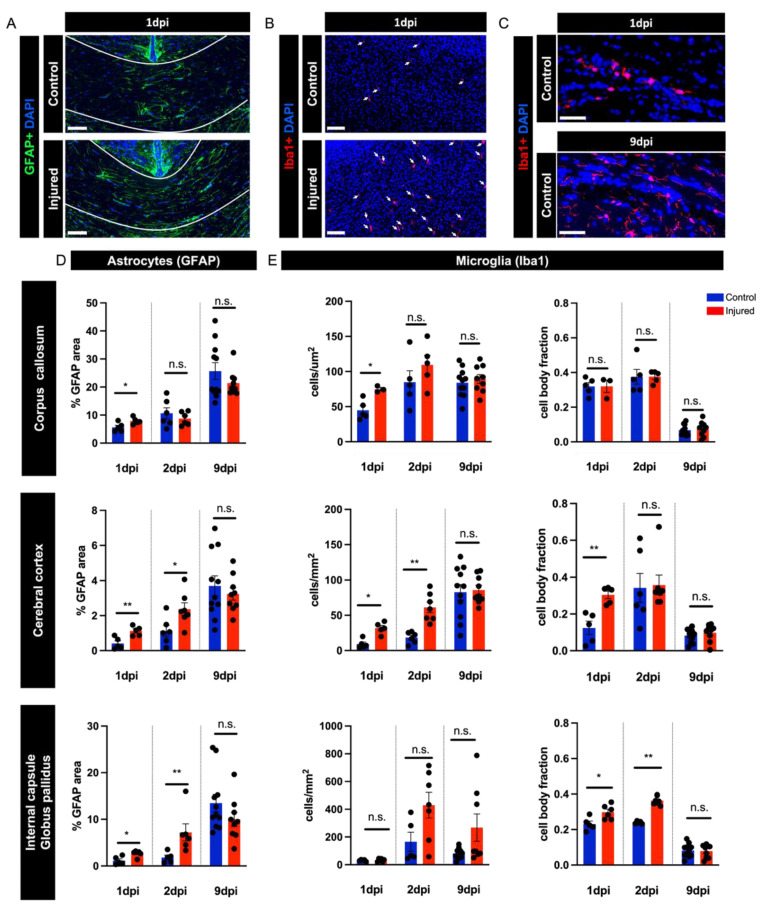
Quantifying reactive changes in astrocytes and microglia. (**A**,**B**) Representative images of control and injured brains at 1 dpi labeled with GFAP in the corpus callosum (**A**) and Iba1 in the cortex (**B**). (**C**) Representative images of the corpus callosum from control mice at 1 dpi and 9 dpi. (**D**) Quantification of GFAP immunohistochemistry in mouse corpus callosum at 1 dpi (U = 3, *p* = 0.0347), 2 dpi (U = 15, *p* = 0.6991), and 9 dpi (U = 44, *p* = 0.7103); in cerebral cortex at 1 dpi (U = 1, *p* = 0.0159), 2 dpi (U = 7, *p* = 0.0513), and 9 dpi (U = 46, *p* = 0.8238); and in internal capsule and globus pallidus at 1 dpi (U = 2, *p* = 0.0173), 2 dpi (U = 1, *p* = 0.0087), and 9 dpi (U = 31, *p* = 0.1754). (**E**) Quantification of Iba1 immunohistochemistry for cells/mm^2^ and cell body fraction (cbf) in mouse corpus callosum at 1 dpi (cells/mm^2^: U = 0, *p* = 0.0357; cbf: U = 7, *p* > 0.999), 2 dpi (cells/mm^2^: U = 6, *p* = 0.222; cbf: U = 9, *p* = 0.5476), and 9 dpi (cells/mm^2^: U = 42, *p* = 0.6027; cbf: U = 45, *p* = 0.7664); in cerebral cortex at 1 dpi (cells/mm^2^: U = 1, *p* = 0.0159; cbf: U =0, *p* = 0.0079), 2 dpi (cells/mm^2^: U = 0, *p* = 0.0012; cbf: U = 14, *p* = 0.3660), and 9 dpi (cells/mm^2^: U = 47, *p* = 0.8820; cbf: U = 34, *p* = 0.2610); and in internal capsule and globus pallidus at 1 dpi (cells/mm^2^: U = 13, *p* = 0.7922; cbf: U = 3, *p* = 0.0303), 2 dpi (cells/mm^2^: U = 5, *p* = 0.0480; cbf: U = 0, *p* = 0.0025); and 9 dpi (cells/mm^2^: U = 31, *p* = 0.3100; cbf: U = 49, *p* > 0.999). Sample amounts: 1 dpi: control *n* = 5, injured *n* = 5; 2 dpi: control *n* = 3–6, injured *n* = 7; 9 dpi: control *n* = 9, injured *n* = 9. Scale bar: 100 µm. Data are presented as mean ± SEM; n.s., not significant; * *p* < 0.05, ** *p* < 0.01.

**Figure 6 biomedicines-10-02810-f006:**
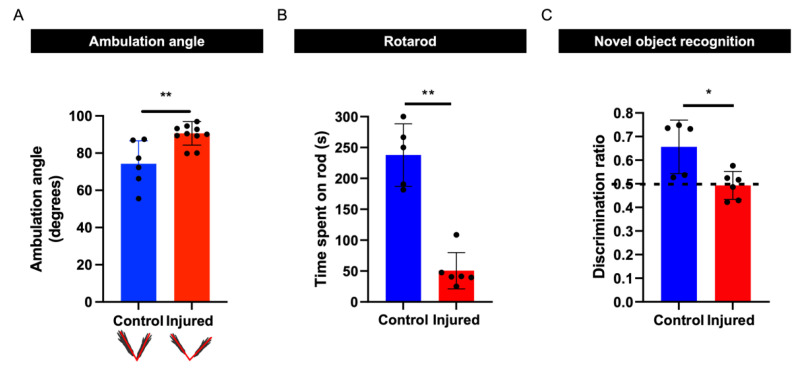
Evaluation of behavior. (**A**) Quantification of hindlimb ambulation angle at 13 dpi, with control *n* = 6 and injured *n* = 11 (U = 4, *p* = 0.0019). (**B**) Quantification of rotarod test, with control *n* = 5 and injured *n* = 6. (**C**) Quantification of novel object recognition task, with *n* = 5 and *n* = 6 (U = 2, *p* = 0.0173). Data are presented as mean ± SEM; * *p* < 0.05 and ** *p* < 0.01.

## Data Availability

The data presented in this study are available on request from the corresponding author.

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
