# Peer review of "A Novel Murine Multi-Hit Model of Perinatal Acute Diffuse White Matter Injury Recapitulates Major Features of Human Disease"

_biomedicines, 2022, doi:10.3390/biomedicines10112810_

Round 1
Reviewer 1 Report
Renz and colleagues characterize a novel murine model of perinatal acute diffuse white matter injury (dWMI) that aims to address the changing epidemiology and reduced severity of dWMI lesions observed in the clinical population. Their multi-hit model captures inflammatory and hypoxic insults in the mouse perinatal period in concordance to insults experienced by premature neonates. This is a relevant and important study. Overall experiments are well designed and clearly presented, however there are some deficiencies that need to be addressed.
1) Based on methods and results it seems that brains were sectioned in the coronal plane at the level of hippocampi. Please provide more details on the AP level that the study was conducted and how this anatomical area was chosen. Please indicate if 1 or more sections and areas were used per mouse for analyses and particularly for the main neuropathological assessment.
2) Different experiments seem to have different number of cases and it is not clear whether different behavioural/histological experiments were performed on the same pups or were independent to each other. Please offer a table documenting experiments and cohorts of mice and whether any (and what) exclusion criteria might have been applied which may explain differences across experiments.
3) For the statistical analysis, authors document that they have assessed for normality and performed either Student t-test (t) or MW tests (U). When presenting results in figures or texts the authors should report full statistics including t, and U and p values e.g t(df)=x, p=y. Most of the comparisons show unequal variances and therefore it is important to state whether Welch correction was performed with t-tests as is appropriate. Finally, experiments with more than two variables (e.g. injury vs time) are typically assessed with 2-way ANOVA/Mixed effect analysis, if possible, in order to establish the contribution of each variable before post-hoc comparisons. The authors should state whether they have adjusted p-values for multiple comparisons or not. The authors should also state whether their design was informed by prior power calculations as several experiments seem underpowered.
4) Fig. 2 and associated text. The authors state that based on TUNEL staining, there is very low frequency of cell death in cortical areas and conclude that their model doesn’t increase cortical cell death. However at 1dpi there is clear trend of nearly 3-fold increase in TUNEL+ cells and the ns is probably because of very low power and great variance in the injury group. While presenting full statistics for such n.s. trends would allow the reader to better appreciate the results presented (e.g. a n.s. two-tailed p value, might correspond to significant one-tailed p-value, depending on prior probabilities) this single experiment and in the absence of further validation (e.g. stereology for cortical neurons) seems inadequate to establish the lack of cortical cell loss in this model. The presence of cortical injury may also be suggested by the prominent microglial reaction.
5) Fig. 4A-C. Figures are heavily annotated with small arrows/arrowheads but explanations are missing in the legend. High-power insets or supplementary images would be helpful in order to indicate what exactly is counted while counting criteria can be presented in methods.
6) While authors assess several aspects of white matter changes, they have not assessed the presence or absence of axonal pathology which would be critical for understanding the severity of the model and the primary or secondary nature of the observed dysmyelination. Parallel sections could be stained with antibodies against APP (and NF) or with Gallyas or De Olmos silver for assessment of axonal pathology
7) Methods stated that control mice were similarly manipulated (e.g. saline injection and left in room air), and therefore the most appropriate designation would be control or sham(-injured) rather than naïve (no manipulation) as is stated across figures/results.
8) In the introduction and discussion of the paper, it is not clear how the authors reached the current design and whether individual insults (LPS or hypoxia) may have similar or different outcomes. For instance systemic injection of LPS is sufficient to cause microglial activation in adult mice.
9) Discuss any weaknesses of the study in the discussion section.
Author Response
Reviewer 1:
Renz and colleagues characterize a novel murine model of perinatal acute diffuse white matter injury (dWMI) that aims to address the changing epidemiology and reduced severity of dWMI lesions observed in the clinical population. Their multi-hit model captures inflammatory and hypoxic insults in the mouse perinatal period in concordance to insults experienced by premature neonates. This is a relevant and important study. Overall experiments are well designed and clearly presented, however there are some deficiencies that need to be addressed.
1) Based on methods and results it seems that brains were sectioned in the coronal plane at the level of hippocampi. Please provide more details on the AP level that the study was conducted and how this anatomical area was chosen. Please indicate if 1 or more sections and areas were used per mouse for analyses and particularly for the main neuropathological assessment.
Thank you for requesting this clarification. Indeed, brains were sectioned in the coronal plane at the level of the hippocampus. This plane was chosen because it allows analysis of the corpus callosum as well as the internal capsule, regions commonly affected in human WMI. The main neuropathological assessment was performed uniquely in this plane. At least 2 brain sections per animal were examined. For IHC experiments, two brain sections were quantified and the results averaged to generate each data point. In rare cases we were only able to quantify IHC results one brain section because of issues with tissue preservation. We have added this information to the Methods section of the paper.
2) Different experiments seem to have different number of cases and it is not clear whether different behavioural/histological experiments were performed on the same pups or were independent to each other. Please offer a table documenting experiments and cohorts of mice and whether any (and what) exclusion criteria might have been applied which may explain differences across experiments.
Thank you for raising this source of confusion. We have included a table documenting each mouse included in the study, the experiments performed on it, and reason for exclusion if excluded. In addition, we have noted exclusion criteria in the Methods section of the manuscript. We did our very best to perform IHC simultaneously on the slides used for each experiment to enable optimal consistency of IHC conditions. In a few cases, a tissue section lost integrity during IHC labeling or was unexpectedly not in the desired plane. These brains were then excluded from the cohort rather than re-stained individually. These instances have been noted.
3) For the statistical analysis, authors document that they have assessed for normality and performed either Student t-test (t) or MW tests (U). When presenting results in figures or texts the authors should report full statistics including t, and U and p values e.g t(df)=x, p=y. Most of the comparisons show unequal variances and therefore it is important to state whether Welch correction was performed with t-tests as is appropriate. Finally, experiments with more than two variables (e.g. injury vs time) are typically assessed with 2-way ANOVA/Mixed effect analysis, if possible, in order to establish the contribution of each variable before post-hoc comparisons. The authors should state whether they have adjusted p-values for multiple comparisons or not. The authors should also state whether their design was informed by prior power calculations as several experiments seem underpowered.
Thank you for this advice. A related point was also raised by Reviewer 3, who suggested that the normality of data cannot be reliably tested at N<30 and that the MW test should therefore be used throughout the manuscript. We have described our MW statistics as requested (U=x, p=y) throughout the text. Although we present data comparing injured and control conditions at multiple timepoints, we have not attempted to make claims regarding the statistical significance of changes in variables over time. For this reason, we have not applied 2-way ANOVA analysis. Power calculations were made based on van Tilborg et al. 2017. These calculations indeed indicate that we do not have significant N to conclude no significant difference between injured and control groups in several cases. We have commented on this point as a weakness of the study. Obtaining the needed N in order to exclude statistical significance in these cases was not feasible within the timeframe of the revision period.
4) Fig. 2 and associated text. The authors state that based on TUNEL staining, there is very low frequency of cell death in cortical areas and conclude that their model doesn’t increase cortical cell death. However at 1dpi there is clear trend of nearly 3-fold increase in TUNEL+ cells and the ns is probably because of very low power and great variance in the injury group. While presenting full statistics for such n.s. trends would allow the reader to better appreciate the results presented (e.g. a n.s. two-tailed p value, might correspond to significant one-tailed p-value, depending on prior probabilities) this single experiment and in the absence of further validation (e.g. stereology for cortical neurons) seems inadequate to establish the lack of cortical cell loss in this model. The presence of cortical injury may also be suggested by the prominent microglial reaction.
Thank you for this point. We agree. In line with Reviewer 3’s request to increase Ns to 5 or more and address power deficiency (as you have) for this experiment in particular, we have quickly generated data on several more brains for this analysis. As you predicted, with this additional data included, we indeed see a significant increase in TUNEL+ cells in the cortex at 1dpi. We have adapted our Results and Discussion section to reflect this finding.
5) Fig. 4A-C. Figures are heavily annotated with small arrows/arrowheads but explanations are missing in the legend. High-power insets or supplementary images would be helpful in order to indicate what exactly is counted while counting criteria can be presented in methods.
Thank you. We have updated the figure legend and included supplementary images to clarify this analysis.
6) While authors assess several aspects of white matter changes, they have not assessed the presence or absence of axonal pathology which would be critical for understanding the severity of the model and the primary or secondary nature of the observed dysmyelination. Parallel sections could be stained with antibodies against APP (and NF) or with Gallyas or De Olmos silver for assessment of axonal pathology
Thank you for raising this point. We have performed density analysis of Neurofilament IHC on the corpus callosum region and included these findings in supplementary data. Given the limited time available for revisions, we were not able to perform APP or Gallyas labeling.
7) Methods stated that control mice were similarly manipulated (e.g. saline injection and left in room air), and therefore the most appropriate designation would be control or sham(-injured) rather than naïve (no manipulation) as is stated across figures/results.
Thank you for pointing this out. We have modified the text as requested.
8) In the introduction and discussion of the paper, it is not clear how the authors reached the current design and whether individual insults (LPS or hypoxia) may have similar or different outcomes. For instance systemic injection of LPS is sufficient to cause microglial activation in adult mice.
Thank you. We have included a more detailed discussion of single and multi-hit models and the logic behind our decision to use both LPS and hypoxia. We did titrate LPS dose and hypoxia in order to achieve a reproducible injury in the absence of excess mortality, but we did not do experiments ourselves to compare outcomes after our individual insults.
9) Discuss any weaknesses of the study in the discussion section.
We now discuss multiple study weaknesses in the discussion section: power limitations, no tissue analysis beyond 11 days post injury, superficial analysis of microglial and astrocyte reactivity.
Reviewer 2 Report
In the present paper, Renz and colleagues describe a novel murine model of perinatal white matter injury, consisting in a two-hit inflammatory-hypoxic paradigm. In this respect, the authors characterize neuronal and demyelinating lesion formation, oligodendrocyte viability and maturation, astrocyte and microglia response, and motor dysfunction following perinatal inflammation/hypoxia, concluding that the proposed model successfully recapitulates key features of human disease.
Overall, the manuscript is well-written and the results described are clear and scientifically sound, making it an helpful addition to the field. However, there are some major points that should be addressed in more detail:
1. The characterization of microglia reactivity is basic and could be easily implemented to provide more information regarding the impact of the proposed injury paradigm on the inflammatory environment. I suggest including a more detailed evaluation of Iba1+ cell morphology (i.e. using skeleton or Scholl analysis or similar tools) and analyzing the extent of Iba1+ co-localization with more functional markers of microglia (i.e. phagocytosis, pro-inflammatory, immunomodulatory). Similarly, the analysis of GFAP staining could be corroborated by evaluating the co-localization with A1/A2 markers.
2. When possible (i.e. figures 4 and 5), it may be more appropriate performing two-way ANOVA instead of t-test, since two variables need to be taken into consideration (naïve/injured ad time after injury). This should allow not only comparing naïve vs injured mice at each time point, but also following the cell dynamics over time after injury.
3. The authors properly mention that perinatal white matter injury manifests as a broad spectrum of motor, cognitive, and behavioral deficits, which are often maintained in adulthood. While the analysis included support the development of motor dysfunction in the proposed model, data regarding the presence of cognitive, memory, or sociability defects are lacking. It would be important to add such information, since pathological conditions associated to perinatal injury in humans are mostly characterized by these defects.
4. I believe that such type of manuscript should include a more deep and detailed discussion comparing the proposed new model to the already utilized ones, highlighting advantages and limitations of this new protocol in the context of state-of-the-art models. This has been partly done regarding some hypoxic/ischemic models, but no mention is done about established models of perinatal inflammatory injury (i.e. using LPS, PolyI:C, IL-1β) or other perinatal injury paradigms.
Author Response
Reviewer 2:
In the present paper, Renz and colleagues describe a novel murine model of perinatal white matter injury, consisting in a two-hit inflammatory-hypoxic paradigm. In this respect, the authors characterize neuronal and demyelinating lesion formation, oligodendrocyte viability and maturation, astrocyte and microglia response, and motor dysfunction following perinatal inflammation/hypoxia, concluding that the proposed model successfully recapitulates key features of human disease.
Overall, the manuscript is well-written and the results described are clear and scientifically sound, making it an helpful addition to the field. However, there are some major points that should be addressed in more detail:
- The characterization of microglia reactivity is basic and could be easily implemented to provide more information regarding the impact of the proposed injury paradigm on the inflammatory environment. I suggest including a more detailed evaluation of Iba1+ cell morphology (i.e. using skeleton or Scholl analysis or similar tools) and analyzing the extent of Iba1+ co-localization with more functional markers of microglia (i.e. phagocytosis, pro-inflammatory, immunomodulatory). Similarly, the analysis of GFAP staining could be corroborated by evaluating the co-localization with A1/A2 markers.
Thank you for raising this point. We agree fully that these are interesting and important lines for further investigation. With regards to microglial morphology, in addition to the cell body fraction metric that we had already provided, we have now analyzed perimeter and Feret’s diameter. We would have expected these parameters to decrease after injury in line with a more ameboid microglial morphology, but instead observe a significant increase in these two metrics across multiple timepoints and brain regions. When we examine images of the analyzed cells, it appears that the cell processes are being excluded from the analysis because they intermittently disappear from the plane of section and that the increase in perimeter and Feret's diameter reflects a larger cell body in the reactive cells rather than increased ramification or cell spread. We have attached this data here for your review instead of including it in the manuscript because we feel that it is misleading in it's current state. We feel that more detailed morphologic analysis of these cells is certainly warranted using confocal imaging and 3D reconstructions combined with Scholl analysis. This analysis was unfortunately not feasible within the time allowed for revision. Functional microglial characterization was unfortunately also not feasible within our revision period, but is also an important future direction. The precise nature of astrocyte reactivity is indeed a fascinating question. Given time limitations, we again felt that this analysis was beyond the scope of the present manuscript. We have added text to the discussion section mentioning these ideas as important next steps.
- When possible (i.e. figures 4 and 5), it may be more appropriate performing two-way ANOVA instead of t-test, since two variables need to be taken into consideration (naïve/injured ad time after injury). This should allow not only comparing naïve vs injured mice at each time point, but also following the cell dynamics over time after injury.
Thank you for raising this point. Indeed, it would be of interest to also examine the significance of differences in our parameters over time. Given limited N with unequal sample variance and uncertain normality (at low N) as well as the necessity for correction for multiple comparisons with ANOVA analysis, we have opted to report only Mann Whitney statistics in this manuscript. We therefore do not make claims about trends in our data over time.
- The authors properly mention that perinatal white matter injury manifests as a broad spectrum of motor, cognitive, and behavioral deficits, which are often maintained in adulthood. While the analysis included support the development of motor dysfunction in the proposed model, data regarding the presence of cognitive, memory, or sociability defects are lacking. It would be important to add such information, since pathological conditions associated to perinatal injury in humans are mostly characterized by these defects.
Agreed! Since submitting our original manuscript, we have conducted novel object testing to complement our motor assay and have observed a significant impairment in recognition memory in injured mice at postnatal day 30. This data is now included in the manuscript. Additional behavioral testing targeting sociability is an important future direction but was not feasible within the revision period.
- I believe that such type of manuscript should include a more deep and detailed discussion comparing the proposed new model to the already utilized ones, highlighting advantages and limitations of this new protocol in the context of state-of-the-art models. This has been partly done regarding some hypoxic/ischemic models, but no mention is done about established models of perinatal inflammatory injury (i.e. using LPS, PolyI:C, IL-1β) or other perinatal injury paradigms.
We have added a detailed discussion of published models (including those involving inflammatory injury) in the discussion section. This discussion is indeed helpful for understanding the added value of our findings in the context of existing models.

Reviewer 3 Report
This work reports on the characterization of a new two-hits perinatal brain injury model in which a the systemic administration of LPS is followed a short hypoxic insult in newborn mice. An immunohistological analysis was performed on different brain areas in order to assess the level of astrocyte and microglial activation, the extent of demyelination and the number of proliferating vs apoptotic oligodendrocytes or oligodendrocyte progenitors. Motor performances were assessed using the rotarod and ambulation angle tests.
This study is overall well performed and the immunohistological analyses are of good quality.
I have concerns regarding the statistical analyses.
1) It is usually admitted that when n are < 30, testing the normality of data is not relevant (even thought tests may give positive results). The statistical analyses should thus be based on the Mann-Whitney test throughout the manuscript
2) The experiments in which n = 3 should be either removed (I prefer a missing data than a potentially misleading data) or completed so to reach a minimum n of 5. This is particularly true for Fig 1B where it is concluded that differences between groups are not significant. Larger n are needed in order to formally conclude that this is indeed a negative result.
Author Response
Reviewer 3:
This work reports on the characterization of a new two-hits perinatal brain injury model in which a the systemic administration of LPS is followed a short hypoxic insult in newborn mice. An immunohistological analysis was performed on different brain areas in order to assess the level of astrocyte and microglial activation, the extent of demyelination and the number of proliferating vs apoptotic oligodendrocytes or oligodendrocyte progenitors. Motor performances were assessed using the rotarod and ambulation angle tests.
This study is overall well performed and the immunohistological analyses are of good quality.
I have concerns regarding the statistical analyses.
1) It is usually admitted that when n are < 30, testing the normality of data is not relevant (even thought tests may give positive results). The statistical analyses should thus be based on the Mann-Whitney test throughout the manuscript
Thank you for this advice. We have revised the statistical analysis throughout the manuscript to be based on the Mann-Whitney test.
2) The experiments in which n = 3 should be either removed (I prefer a missing data than a potentially misleading data) or completed so to reach a minimum n of 5. This is particularly true for Fig 1B where it is concluded that differences between groups are not significant. Larger n are needed in order to formally conclude that this is indeed a negative result.
Thank you for this comment. The limited power of several experiments is an important limitation of our study. We have worked hard during the 10 days allotted for revisions to increase our animal numbers to a minimum of 5 for each experiment. In the case of Figure 1B, this resulted in the emergence of statistical significance. Only in the case of microglia cell number and reactivity index at 1dpi were we unable to achieve N=5. The differences that we observed at this datapoint were statistically significant. We have added a discussion of limited power and thus inability to formally rule out statistical significance to the discussion section of the manuscript.
Round 2
Reviewer 1 Report
Authors have addressed all concerns and significantly improved the manuscript, by providing new data, clarifications or by noting limitations of data in the discussion.
Reviewer 2 Report
The authors properly addressed most of the issues raised by the reviewer.
Look forward to seeing future research investigating microglia and astrocyte responses in this new model.
Reviewer 3 Report
The authors adressed the issues raised in review round 1